# Point Based Value Iteration with Optimal Belief Compression for Dec-POMDPs

**Liam MacDermed**
College of Computing
Georgia Institute of Technology
Atlanta, GA 30332
liam@cc.gatech.edu

**Charles L. Isbell**
College of Computing
Georgia Institute of Technology
Atlanta, GA 30332
isbell@cc.gatech.edu

## Abstract

We present four major results towards solving decentralized partially observable Markov decision problems (DecPOMDPs) culminating in an algorithm that outperforms all existing algorithms on all but one standard infinite-horizon benchmark problems. (1) We give an integer program that solves collaborative Bayesian games (CBGs). The program is notable because its linear relaxation is very often integral. (2) We show that a DecPOMDP with bounded belief can be converted to a POMDP (albeit with actions exponential in the number of beliefs). These actions correspond to strategies of a CBG. (3) We present a method to transform any DecPOMDP into a DecPOMDP with bounded beliefs (the number of beliefs is a free parameter) using optimal (not lossless) belief compression. (4) We show that the combination of these results opens the door for new classes of DecPOMDP algorithms based on previous POMDP algorithms. We choose one such algorithm, point-based valued iteration, and modify it to produce the first tractable value iteration method for DecPOMDPs that outperforms existing algorithms.

## 1 Introduction

Decentralized partially observable Markov decision processes (DecPOMDPs) are a popular model for cooperative multi-agent decision problems; however, they are NEXP-complete to solve [15]. Unlike single agent POMDPs, DecPOMDPs suffer from a doubly-exponential curse of history [16]. Not only do agents have to reason about the observations they see, but also about the *possible* observations of other agents. This causes agents to view their world as non-Markovian because even if an agent returns to the same underlying state of the world, the dynamics of the world may appear to change due to other agent's holding different beliefs and taking different actions. Also, for POMDPs, a sufficient belief space is the set of probability distributions over possible states. In the case of DecPOMDPs an agent must reason about the beliefs of other agents (who are recursively reasoning about beliefs as well), leading to *nested beliefs* that can make it impossible to losslessly reduce an agent's knowledge to less than its full observation history.

This lack of a compact belief-space has prevented value-based dynamic programming methods from being used to solve DecPOMDPs. While value methods have been quite successful at solving POMDPs, all current DecPOMDP approaches are policy-based methods, where policies are sequentially improved and evaluated at each iteration. Even using policy methods, the curse of history is still a big problem, and current methods deal with it in a number of different ways. [5] simply removed beliefs with low probability. Some use heuristics to prune (or never explore) particular belief regions [19, 17, 12, 11]. Other approaches merge beliefs together (*i.e.,* belief compression) [5, 6]. This can sometimes be done losslessly [13], but such methods have limited applicability and still usually result in an exponential explosion of beliefs. There have also been approaches that attempt to operate directly on the infinitely nested belief structure [4], but these are approximations

of unknown accuracy (if we stop at the $n^{th}$ nested belief the $n^{th} + 1$ could dramatically change the outcome). All of these approaches have gotten reasonable empirical results in a few limited domains but ultimately scale and generalize poorly.

Our solution to the curse of history is simple: to assume that it doesn't exist, or more precisely, that the number of possible beliefs at any point in time is bounded. While simple, the consequences of this assumption turn out to be quite powerful: a bounded-belief DecPOMDP can be converted into an equivalent POMDP. This conversion is accomplished by viewing the problem as a sequence of cooperative Bayesian Games (CBGs). While this view is well established, our use of it is novel. We give an efficient method for solving these CBGs and show that any DecPOMDP can be accurately approximated by a DecPOMDP with bounded beliefs. These results enable us to utilize existing POMDP algorithms, which we explore by modifying the PERSEUS algorithm [18]. Our resulting algorithm is the first true value-iteration algorithm for DecPOMDPs (where no policy information need be retained from iteration to iteration) and outperforms existing algorithms.

## 2 DecPOMDPs as a sequence of cooperative Bayesian Games

Many current approaches for solving DecPOMDPs view the decision problem faced by agents as a sequence of CBGs [5]. This view arises from first noting that a complete policy must prescribe an action for every belief-state of an agent. This can take the form of a strategy (a mapping from belief to action) for each time-step; therefore at each time-step agents must choose strategies such that the expectation over their joint-actions and beliefs maximizes the sum of their immediate reward combined with the utility of their continuation policy. This decision problem is equivalent to a CBG.

Formally, we define a DecPOMDP as the tuple $\langle N, A, S, O, P, R, s^{(0)} \rangle$ where: $N$ is the set of $n$ players. $A = \prod_{i=1}^{n} A_i$ is the set of joint-actions. $S$ is the set of states. $O = \prod_{i=1}^{n} O_i$ is the set of joint-observations. $P : S \times A \to \Delta(S \times O)$ is the probability transition function with $P(s', \vec{o}|s, \vec{a})$ being the probability of ending up in state $s'$ with observations $\vec{o}$ after taking joint-action $\vec{a}$ in state $s$. $R : S \times A \to \mathbb{R}$ is the shared reward function. And $s^{(0)} \in \Delta(S)$ is the initial state distribution.

The CBG consists of a common-knowledge distribution over all possible joint-beliefs along with a reward for each joint-action/belief. Naively, each observation history corresponds to a belief. If beliefs are more compact (*i.e.,* through belief compression), then multiple histories can correspond to the same belief. The joint-belief distribution is commonly known because it depends only on the initial state and player's policies, which are both commonly known (due to common planning or rationality). These beliefs are the types of the Bayesian game. We can compute the current common-knowledge distribution (without belief compression) recursively: The probability that joint-type $\theta^{t+1} = \langle \vec{o}, \theta^t \rangle$ at time $t+1$ is given by: $\tau^{t+1}(\langle \vec{o}, \theta^t \rangle) = \sum_{s^{t+1}} Pr[s^{t+1}, \langle \vec{o}, \theta^t \rangle]$ where:

$$Pr[s^{t+1}, \langle \vec{o}, \theta^t \rangle] = \sum_{s^t} \sum_{\theta^t} P(s^{t+1}, \vec{o}|s^t, \pi_{\theta^t}) Pr[s^t, \theta^t] \tag{1}$$

The actions of the Bayesian game are the same as in the DecPOMDP. The rewards to the Bayesian game are ideally the immediate reward $R = \sum_{s^t} \sum_{\theta^t} R(s^t, \pi_{\theta^t}) Pr[s^t, \theta^t]$ along with the utility of the best continuation policy. However, knowing the best utility is tantamount to solving the problem. Instead, an estimation can be used. Current approaches estimate the value of the DecPOMDP as if it were an MDP [19], a POMDP [17], or with delayed communication [12]. In each case, the solution to the Bayesian game is used as a heuristic to guide policy search.

## 3 An Integer Program for Solving Collaborative Bayesian Games

Many DecPOMDP algorithms use the idea that a DecPOMDP can be viewed as a sequence of CBGs to divide policy optimization into smaller sub-problems; however, solving CBGs themselves is NP-complete [15]. Previous approaches have solved Bayesian games by enumerating all strategies, with iterated best response (which is only locally optimal) or branch and bound search [10]. Here we present a novel integer linear program that solves for an optimal pure strategy Bayes-Nash equilibrium (which always exists for games with common payoffs). While integer programming is still NP-complete, our formulation has a huge advantage: the linear relaxation is itself a correlated communication equilibrium [7], empirically very often integral (above 98% for our experiments in section 7). This allows us to optimally solve our Bayesian games very efficiently.

Our integer linear program for Bayesian game $\langle N, A, \Theta, \tau, R \rangle$ optimizes over Boolean variables $x_{\vec{a},\theta}$, one for each joint-action for each joint-type. Each variable represents the probability of joint-action $\vec{a}$ being taken if the agent's types are $\theta$. Constraints must be imposed on these variables to insure that they form a proper probability distribution and that from each agent's perspective, its action is conditionally independent of other agents' types.

These restrictions can be expressed by the following linear constraints (equation (2)):

$$
\boxed{
\begin{array}{c}
\text{For each agent } i, \text{ joint-type } \theta, \text{ and partial joint-actions of other agents } \vec{a}_{-i} \\[2mm]
\displaystyle\sum_{a_i \in A_i} x_{\vec{a},\theta} \;=\; x_{\langle \vec{a}_{-i}, \theta_{-i} \rangle} \\[4mm]
\text{for each } \theta \in \Theta: \displaystyle\sum_{\vec{a} \in A} x_{\vec{a},\theta} = 1 \quad \text{and} \quad \text{for each } \theta \in \Theta, \; \vec{a} \in A: \; x_{\vec{a},\theta} \geq 0
\end{array}
}
\tag{2}
$$

In order to make the description of the conditional independence constraints more concise, we use the additional variables $x_{\langle \vec{a}_{-i}, \theta_{-i} \rangle}$. These can be immediately substituted out. These represent the posterior probability that agent $i$, after becoming type $\theta_i$, thinks other agents will take actions $\vec{a}_{-i}$ when having types $\theta_{-i}$. These constraints enforce that an agent's posterior probabilities are unaffected by other agent's observations. Any feasible assignment of variables $x_{\vec{a},\theta}$ represents a valid agent-normal-form correlated equilibria (ANFCE) strategy for the agents, and any integral solution is a valid pure strategy BNE. In order to find the optimal solution for a game with distribution over types $\tau \in \Delta(\Theta)$ and rewards $R : \Theta \times A \to \mathbb{R}^n$ we can solve the integer program: Maximize $\sum \tau_\theta R(\theta, \vec{a}) x_{\vec{a},\theta}$ over variables $x_{\vec{a},\theta} \in \{0, 1\}$ subject to constraints (2).

An ANFCE generalizes Bayes-Nash equilibria: a pure strategy ANFCE is a BNE. We can view ANFCEs as having a mediator that each agent tells its type to and receives an action recommendation from. An ANFCE is then a probability distribution across joint type/actions such that agents do not want to lie to the mediator nor deviate from the mediator's recommendation. More importantly, they cannot deduce any information about other agent's types from the mediator's recommendation. We cannot use an ANFCE directly, because it requires communication; however, a deterministic (*i.e.,* integral) ANFCE requires no communication and is a BNE.

## 4 Bounded Belief DecPOMDPs

Here we show that we can convert a bounded belief DecPOMDP (BB-DecPOMDP) into an equivalent POMDP (that we call the *belief-POMDP*). A BB-DecPOMDP is a DecPOMDP where each agent $i$ has a fixed upper bound $|\Theta_i|$ for the number of beliefs at each time-step. The belief-POMDP's states are factored, containing each agent's belief along with the DecPOMDP's state. The POMDP's actions are joint-strategies. Recently, Dibangoye *et al.* [3] showed that a finite horizon DecPOMDP can be converted into a finite horizon POMDP where a probability distribution over histories is a sufficient statistic that can be used as the POMDP's state. We extend this result to infinite horizon problems when beliefs are bounded (note that a finite horizon problem always has bounded belief). The main insight here is that we do not have to remember histories, only a distribution over belief-labels (without any a priori connection to the belief itself) as a sufficient statistic. As such the same POMDP states can be used for all time-steps, enabling infinite horizon problems to be solved.

In order to create the belief-POMDP we first transform observations so that they correspond one-to-one with beliefs for each agent. This can be achieved naively by folding the previous belief into the new observation so that each agent receives a [previous-belief, observation] pair; however, because an agent has at most $|\Theta_i|$ beliefs we can partition these histories into at most $|\Theta_i|$ information-equivalent groups. Each group corresponds to a distinct belief and instead of the [previous-belief, observation] pair we only need to provide the new belief's label.

Second, we factor our state to include each agent's observation (now a belief-label) along with the original underlying state. This transformation increases the state space polynomially. Third, recall that a belief is the sum of information that an agent uses to make decisions. If agents know each other's policies (*e.g.,* by constructing a distributed policy together) then our modified state (which includes beliefs for each agent) fully determines the dynamics of the system. States now

appear Markovian again. Therefore, a probability distribution across states is once again a sufficient plan-time statistic (as proven in [3] and [9]). This distribution exactly corresponds to the Bayesian prior (after receiving the belief-observation) of the common knowledge distribution of the current Bayesian game being played as given by equation (1).

Finally, its important to note that beliefs do not directly affect rewards or transitions. They therefore have no meaning beyond the prior distribution they induce. We can therefore freely relabel and reorder beliefs without changing the decision problem. This allows belief-observations in one time-step to use the same observation labels in the next time-step, even if the beliefs are different (in which case the distribution will be different). We can use this fact to fold our mapping from histories to belief-labels into the belief-POMDP's transition function.

We now formally define the belief-POMDP $\langle A', S', O', P', R', s'^{(0)} \rangle$ converted from BB-DecPOMDP $\langle N, A, S, O, P, R, s^{(0)} \rangle$ (with belief labels $\Theta_i$ for each agent). The belief-POMDP has factored states $\langle \omega, \theta_1, \cdots, \theta_n \rangle \in S'$ where $\omega \in S$ is the underlying state and $\theta_i \in \Theta_i$ is agent $i$'s belief. $O' = \{\}$ (no observations). $A' = \prod_{i=1}^{n} \prod_{j=1}^{|\Theta_i|} A_i$ is the set of actions (one action for each agent for each belief). $P'(s'|s, a), = \sum_{[\theta, o] = \theta'} P(\omega', o | \omega, \langle a_{\theta_1}, \cdots, a_{\theta_n} \rangle)$ (a sum over equivalent joint-beliefs) where $a_{\theta_i}$ is the action agent $i$ would take if holding belief $\theta_i$. $R'(s, a) = R(\omega, \langle a_{\theta_1}, \cdots, a_{\theta_n} \rangle)$ and $s'^{(0)}_\omega = s^{(0)}$ is the initial state distribution with each agent having the same belief.

Actions in this belief-POMDP are pure strategies for each agent specifying what each agent should do for every belief they might have. In other words it is a mapping from observation to action ($A' = \{\Theta \to A\}$). The action space thus has size $\prod_i |A_i|^{|\Theta_i|}$ which is exponentially more actions than the number of joint-actions in the BB-DecPOMDP. Both the transition and reward functions use the modified joint-action $\langle a_{\theta_1}, \cdots, a_{\theta_n} \rangle$ which is the action that would be taken once agents see their beliefs and follow action $a \in A$. This makes the single agent in the belief-POMDP act like a centralized mediator playing the sequence of Bayesian games induced by the BB-DecPOMDP. At every time-step this centralized mediator must give a strategy to each agent (a solution to the current Bayesian game). The mediator only knows what is commonly known and thus receives no observations.

This belief-POMDP is decision equivalent to the BB-DecPOMDP. The two models induce the same sequence of CBGs; therefore, there is a natural one-to-one mapping between polices of the two models that yield identical utilities. We show this constructively by providing the mapping:

**Lemma 4.1.** *Given BB-DecPOMDP $\langle N, A, O, \Theta, S, P, R, s^{(0)} \rangle$ with policy $\pi : \Delta(S) \times O \to A$ and belief-POMDP $\langle A', S', O', P', R', s'^{(0)} \rangle$ as defined above, with policy $\pi' : \Delta(S') \to \{O' \to A'\}$ then if $\pi(s^{(t)})_{i,o} = \pi'_i(s^{(t)}, o)$, then $V_\pi(s^{(0)}) = V'_{\pi'}(s'^{(0)})$ (the expected utility of the two policies are equal).*

We have shown that BB-DecPOMDPs can be turned into POMDPs but this does not mean that we can easily solve these POMDPs using existing methods. The action-space of the belief-POMDP is exponential with respect to the number of observations of the BB-DecPOMDP. Most existing POMDP algorithms assume that actions can be enumerated efficiently, which isn't possible beyond the simplest belief-POMDP. One notable exception is an extension to PERSEUS that randomly samples actions [18]. This approach works for some domains; however, often for decentralized problems only one particular strategy proves effective, making a randomized approach less useful. Luckily, the optimal strategy is the solution to a CBG, which we have already shown how to solve. We can then use existing POMDP algorithms and replace action maximization with an integer linear program. We show how to do this for PERSEUS below. First, we make this result more useful by giving a method to convert a DecPOMDP into a BB-DecPOMDP.

## 5 Optimal belief compression

We present here a novel and optimal belief compression method that transforms any DecPOMDP into a BB-DecPOMDP. The idea is to let agents themselves decide how they want to merge their beliefs and to add this decision directly into the problem's structure. This pushes the onus of belief compression onto the BB-DecPOMDP solver instead of an explicit approximation method. We give agents the ability to optimally compress their own beliefs by interleaving each normal time-step

(where we fully expand each belief) with a compression time-step (where the agents must explicitly decide how to best merge beliefs). We call these phases belief expansion and belief compression, respectively.

The first phase acts like the original DecPOMDP without any belief compression: the observation given to each agent is its previous belief along with the DecPOMDP's observation. No information is lost during this phase; each observation for each agent-type (agents holding the same belief are the same type) results in a distinct belief. This belief expansion occurs with the same transitions and rewards as the original DecPOMDP.

The dynamics of the second phase are unrelated to the DecPOMDP. Instead, an agent's actions are decisions about how to compress its belief. In this phase, each agent-type must choose its next belief but they only have a fixed number of beliefs to choose from (the number of beliefs $t_i$ is a free parameter). All agent-types that choose the same belief will be unable to distinguish themselves in the next time-step; the belief label in the next time-step will equal the action index they take in the belief compression phase. All rewards are zero. This second phase can be seen as a purely mental phase and does not affect the environment beyond changes to beliefs although as a technical matter, we convert our discount factor to its square root to account for these new interleaved states.

Given a DecPOMDP $\langle N, A, S, O, P, R, s^{(0)} \rangle$ (with states $\omega \in S$ and observations $\sigma \in O$) we formally define the BB-DecPOMDP approximation model $\langle N', A', O', S', P', R', s'^{(0)} \rangle$ with belief set size parameters $t_1, \cdots, t_n$ as:

- $N' = N$ and $A'_i = \{a_1, \cdots, a_{\max(|A_i|, \ t_i)}\}$
- $O'_i = \{O_i, \emptyset\} \bigtimes \{1, 2, \cdots, t_i\}$ with factored observation $o_i = \langle \sigma_i, \theta_i \rangle \in O'$
- $S' = S \bigtimes O'$ with factored state $s = \langle \omega, \sigma_1, \theta_1, \cdots, \sigma_n, \theta_n \rangle \in S'$.
- $P'(s'|s, a) = \begin{cases} P(\omega', \langle \sigma'_1, \cdots, \sigma'_n \rangle | \omega, a) & \text{if } \forall i : \sigma_i = \emptyset, \sigma'_i \neq \emptyset \text{ and } \theta'_i = \theta_i \\ 1 & \text{if } \forall i : \sigma_i \neq \emptyset, \sigma'_i = \emptyset \text{ and } \theta'_i = a_i, \omega' = \omega \\ 0 & \text{otherwise} \end{cases}$
- $R'(s, a) = \begin{cases} R(\omega, a) & \text{if } \forall i : \sigma_i = \emptyset \\ 0 & \text{otherwise} \end{cases}$
- $s'^{(0)} = \langle s^{(0)}, \emptyset_1, 1, \cdots, \emptyset_n, 1 \rangle$ is the initial state distribution

We have constructed the BB-DecPOMDP such that at each time-step agents receive two observations: an observation factor $\sigma$, and their belief factor $\theta$ (*i.e.,* type). The observation factor is $\emptyset$ at the expansion phase and the most recent observation, as given by the DecPOMDP, when starting the compression phase. The observation factor therefore distinguishes which phase the model is currently in. Agents should either all have $\sigma = \emptyset$ or none of them. The probability of transitioning to a state where some agents have the empty set observation while others don't is always zero. Note that transitions during the contraction phase are deterministic (probability one) and the underlying state $\omega$ does not change. The action set sizes may be different in the two phases, however we can easily get around this problem by mapping any action outside of the designated actions to an equivalent one inside the designated action. The new BB-DecPOMDP's state size is $|S'| = |S|(|O| + 1)^n t^n$.

## 6  Point-based value iteration for BB-DecPOMDPs

We have now shown that a DecPOMDP can be approximated by a BB-DecPOMDP (using optimal belief-compression) and that this BB-DecPOMDP can be converted into a belief-POMDP where selecting an optimal action is equivalent to solving a collaborative Bayesian game (CBG). We have given a (relatively) efficient integer linear program for solving these CBG. The combination of these three results opens the door for new classes of DecPOMDP algorithms based on previous POMDP algorithms. The only difference between existing POMDP algorithms and one tailored for BB-DecPOMDPs is that instead of maximizing over actions (which are exponential in the belief-POMDP), we must solve a stage-game CBG equivalent to the stage decision problem.

Here, we develop an algorithm for DecPOMDPs based on the PERSEUS algorithm [18] for POMDPs, a specific version of point-based value iteration (PBVI) [16]. Our value function representation is a standard convex and piecewise linear value-vector representation over the belief

simplex. This is the same representation that PERSEUS and most other value based POMDP algorithms use. It consists of a set of hyperplanes $\Gamma = \{\alpha_1, \alpha_2, \cdots, \alpha_m\}$ where $\alpha_i \in \mathbb{R}^{|S|}$. These hyperplanes each represent the value of a particular policy across beliefs. The value function is then the maximum over all hyperplanes. For a belief $b \in \mathbb{R}^{|S|}$ its value as given by the value function $\Gamma$ is $V_\Gamma(b) = \max_{\alpha \in \Gamma} \alpha \cdot b$. Such a representation acts as both a value function and an implicit policy. While each $\alpha$ vector corresponds to the value achieved by following an unspecified policy, we can reconstruct that policy by computing the best one-step strategy, computing the successor state, and repeating the process.

The high-level outline of our point-based algorithm is the same as PERSEUS. First, we sample common-knowledge beliefs and collect them into a belief set $B$. This is done by taking random actions from a given starting belief and recording the resulting belief states. We then start with a poor approximation of the value function and improve it over successive iterations by performing a one-step backup for each belief $b \in B$. Each backup produces a policy which yields a value-vector to improve the value function. PERSEUS improves standard PBVI during each iteration by skipping beliefs already improved by another backup. This reduces the number of backups needed. In order to operate on belief-POMDPs, we replace PERSEUS' backup operation with one that uses our integer program.

In order to backup a particular belief point we must maximize the utility of a strategy $x$. The utility is computed using the immediate reward combined with our value-function's current estimate of a chosen continuation policy that has value vector $\alpha$. Thus, a resulting belief $b'$ will achieve estimated value $\sum_s b'(s)\alpha(s)$. The resulting belief $b'$ after taking action $\vec{a}$ from belief $b$ is $b'(s') = \sum_s b(s)P(s'|s,\vec{a})$. Putting these together, along with the probabilities $x_{\vec{a},s}$ of taking action $\vec{a}$ in state $s$ we get the value of a strategy $x$ from belief $b$ followed by continuation utility $\alpha$:

$$V_{\vec{a},\alpha}(b) = \sum_{s \in S} b(s) \sum_{\vec{a} \in A} \sum_{s' \in S} P(s'|s,\vec{a})\alpha(s')x_{\vec{a},s} \qquad (3)$$

This is the quantity that we wish to maximize, and can combine with constraints (2) to form an integer linear program that returns the best action for each agent for each observation (strategy) given a continuation policy $\alpha$. To find the best strategy/continuation policy pair, we can perform this search over all continuation vectors in $\Gamma$:

| | |
|---|---|
| **Maximize:** | equation 3 |
| **Over:** | $x \in \{0,1\}^{|S||A|}$, $\alpha \in \Gamma$ |
| **Subject to:** | inequalities 2 |

$\qquad (4)$

Each integer program has $|S||A|$ variables but for underlying state factors (nature's type) there are only $|\Theta||A|$ linearly independent constraints - one for each unobserved factor and joint-action. Therefore the number of unobserved states does not increase the number of free variables. Taking this into account, the number of free variables is $\mathbb{O}(|O||A|)$. The optimization problem given above requires searching over all $\alpha \in \Gamma$ to find the best continuation value-vector. We could solve this problem as one large linear program with a different set of variables for each $\alpha$, however each set of variables would be independent and thus can be solved faster as separate individual problems.

We initialize our starting value function $\Gamma^0$ to have a single low conservative value vector (such as $\{R_{\min}/\gamma)\}^n$). Every iteration then attempts to improve the value function at each belief in our belief set $B$. A random common-knowledge belief $b \in B$ is selected and we compute an improved policy for that belief by performing a one step backup. This backup involves finding the best immediate strategy-profile (an action for each observation of each agent) at belief $b$ along with the best continuation policy from $\Gamma^t$. We then compute the value of the resulting strategy + continuation policy (which is itself a policy) and insert this new $\alpha$-vector into $\Gamma^{t+1}$. Any belief that is improved by $\alpha$ (including $b$) is removed from $B$. We then select a new common-knowledge belief and iterate until every belief in $B$ has been improved. We give this as algorithm 1.

This algorithm will iteratively improve the value function at all beliefs. The algorithm stops when the value function improves less than the stopping criterion $\epsilon_\Gamma$. Therefore, at every iteration at least one of the beliefs must improve by at least $\epsilon_\Gamma$. Because the value function at every belief is bounded

**Algorithm 1** The modified point-based value iteration for DecPOMDPs

| | |
|---|---|
| Inputs: | DecPOMDP $M$, discount $\gamma$, belief bounds $|\Theta_i|$, stopping criterion $\epsilon_\Gamma$ |
| Output: | value function $\Gamma$ |

1: $\langle N, A, O, S, P, R, s^{(0)} \rangle \Leftarrow$ BB-DecPOMDP approximation of $M$ as described in section 5
2: $B^\forall \Leftarrow$ sampling of states using a random walk from $s^{(0)}$
3: $\Gamma' \Leftarrow \{\langle R_{\min}/\gamma, \cdots, R_{\min}/\gamma \rangle\}$
4: **repeat**
5:     $B \Leftarrow B^\forall; \quad \Gamma \Leftarrow \Gamma'; \quad \Gamma' \Leftarrow \emptyset$
6:     **while** $B \neq \emptyset$ **do**
7:         $b \Leftarrow \text{Rand}(b \in B)$
8:         $\alpha \Leftarrow \Gamma(b)$
9:         $\alpha' \Leftarrow$ optimal point of integer program (4)
10:        **if** $\alpha'(b) > \alpha(b)$ **then**
11:            $\alpha \Leftarrow \alpha'$
12:        $\Gamma' \Leftarrow \Gamma' \bigcup \alpha$
13:        **for all** $b \in B$ **do**
14:            **if** $\alpha(b) > \Gamma(b)$ **then**
15:                $B \Leftarrow B/b$
16: **until** $\Gamma' - \Gamma < \epsilon_\Gamma$
17: **return** $\Gamma$

above by $R_{\max}/\gamma$ we can guarantee that the algorithm will take fewer than $(|B|R_{\max})/(\gamma \cdot \epsilon_\Gamma)$ iterations.

Algorithm 1 returns a value function. Ultimately we want a policy. Using the value function a policy can be constructed in a greedy manner for each player. This is accomplished using a very similar procedure to how we construct a policy greedily in the fully observable case. Every time-step the actors in the world can dynamically compute their next action without needing to plan their entire policy.

## 7   Experiments

We tested our algorithm on six well known benchmark problems [1, 14]: DecTiger, Broadcast, Grid-small, Cooperative Box Pushing, Recycling Robots, and Wireless Network. On all of these problems we met or exceeded the current best solution. This is particularly impressive considering that some of the algorithms were designed to take advantage of specific problem structure, while our algorithm is general. We also attempted to solve the Mars Rovers problem, except its belief-POMDP transition model was too large for our 8GB memory limit.

We implemented our PBVI for BB-DecPOMDP algorithm in Java using the GNU Linear Programming Kit to solve our integer programs. We ran the algorithm on all six benchmark problems using the dynamic belief compression approximation scheme to convert each of the DecPOMDP problems into BB-DecPOMDPs. For each problem we converted them into a BB-DecPOMDP with one, two, three, four, and five dynamic beliefs (the value of $t_i$).

We used the following fixed parameters while running the PBVI algorithm: $\epsilon_\Gamma = 0.0005$. We sampled 3,000 belief points to a maximum depth of 36. All of the problems, except Wireless, were solved using a discount factor $\gamma$ of 0.9 for the original problem and $\sqrt{0.9}$ for our dynamic approximation (recall that an agent visits two states for every one of the original problem). Wireless has a discount factor of 0.99. To compensate for this low discount factor in this domain, we sampled 30,000 beliefs to a depth of 360. Our empirical evaluations were run on a six-core 3.20GHz Phenom processor with 8GB of memory. We terminated the algorithm and used the current value if it ran longer than a day (only Box Pushing and Wireless took longer than five hours). The final value reported is the value of the computed decentralized policy on the original DecPOMDP run to a horizon which pushed the utility error below the reported precision.

Our algorithms performed very well on all benchmark problems (table 7). Surprisingly, most of the benchmark problems only require two approximating beliefs in order to beat the previously best

| | $|S|$ | $|A_i|$ | $|O_i|$ | Previous Best Utility | 1-Belief Utility | $|\Gamma|$ | 2-Beliefs Utility | $|\Gamma|$ |
|---|---|---|---|---|---|---|---|---|
| Dec-Tiger | 2 | 2 | 2 | 13.4486 [14] | -20.000 | 2 | 4.6161 | 187 |
| Broadcast | 4 | 2 | 2 | 9.1 [1] | 9.2710 | 36 | 9.2710 | 44 |
| Recycling | 4 | 3 | 2 | 31.92865 [1] | 26.3158 | 8 | 31.9291 | 13 |
| Grid small | 16 | 5 | 2 | 6.89 [14] | 5.2716 | 168 | 6.8423 | 206 |
| Box Pushing | 100 | 4 | 5 | 149.854 [2] | 127.1572 | 258 | 223.8674 | 357 |
| Wireless | 64 | 2 | 6 | -175.40 [8] | -208.0437 | 99 | -167.1025 | 374 |

| | 3-Beliefs Utility | $|\Gamma|$ | 4-Beliefs Utility | $|\Gamma|$ | 5-Beliefs Utility | $|\Gamma|$ |
|---|---|---|---|---|---|---|
| Dec-Tiger | 13.4486 | 231 | 13.4486 | 801 | 13.4486 | 809 |
| Broadcast | 9.2710 | 75 | 9.2710 | 33 | 9.2710 | 123 |
| Recycling | 31.9291 | 37 | 31.9291 | 498 | 31.9291 | 850 |
| Grid small | 6.9826 | 276 | 6.9896 | 358 | 6.9958 | 693 |
| Box Pushing | 224.1387 | 305 | - | - | - | - |

Table 1: Utility achieved by our PBVI-BB-DecPOMDP algorithm compared to the previously best known policies on a series of standard benchmarks. Higher is better. Our algorithm beats all previous results except on Dec-Tiger where we believe an optimal policy has already been found.

known solution. Only Dec-Tiger needs three beliefs. None of the problems benefited substantially from using four or five beliefs. Only grid-small continued to improve slightly when given more beliefs. This lack of improvement with extra beliefs is strong evidence that our BB-DecPOMDP approximation is quite powerful and that the policies found are near optimal. It also suggests that these problems do not have terribly complicated optimal policies and new benchmark problems should be proposed that require a richer belief set.

The belief-POMDP state-space size is the primary bottleneck of our algorithm. Recall that this state-space is factored causing its size to be $O(|S||O|^n|t|^n)$. This number can easily become intractably large for problems with a moderate number of states and observations, such as the Mars Rovers problem. Taking advantage of sparsity can mitigate this problem (our implementation uses sparse vectors), however value-vectors tend to be dense and thus sparsity is only a partial solution. A large state-space also requires a greater number of belief samples to adequately cover and represent the value-function; with more states it becomes increasingly likely that a random walk will fail to traverse a desirable region of the state-space. This problem is not nearly as bad as it would be for a normal POMDP because much of the belief-space is unreachable and a belief-POMDP's value function has a great deal of symmetry due to the label invariance of beliefs (a relabeling of beliefs will still have the same utility).

# 8 Conclusion

This paper presented three relatively independent contributions towards solving DecPOMDPs. First, we introduce an efficient integer program for solving collaborative Bayesian games. Other approaches require solving CBGs as a sub-problem, and this could directly improve those algorithms. Second, we showed how a DecPOMDP with bounded belief can be converted into a POMDP. Almost all methods bound the beliefs in some way (through belief compression or finite horizons), and viewing these problems as POMDPs with large action spaces could precipitate new approaches. Third, we showed how to achieve optimal belief compression by allowing agents themselves to decide how best to merge beliefs. This allows any DecPOMDP to be converted into a BB-DecPOMDP. Finally, these independent contributions can be combined together to permit existing POMDP algorithm (here we choose PERSEUS) to be used to solve DecPOMDPs. We showed that this approach is a significant improvement over existing infinite-horizon algorithms. We believe this opens the door towards a large and fruitful line of research into modifying and adapting existing value-based POMDP algorithms towards the specific difficulties of belief-POMDPs.

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
