[Reviews · NeurIPS 2013]

Submitted by Assigned_Reviewer_3

SUMMARY:

This paper proposes a new method for Dec-POMDP planning that is built out of several components. The first is a new way of solving cooperative Bayesian games using an integer linear program. The second is the transformation of the Dec-POMDP to a belief POMDP in which a "centralized mediator" must select at each timestep the best action for each agent-belief pair. The third is to automate the discovery of optimal belief compression by dividing each timestep into two parts, the first corresponding to the original Dec-POMDP and the second giving each agent a chance to select how its beliefs in that timestep are mapped to a bounded set and thus compressed. The fourth assembles these components together into a point-based value iteration method that solves the resulting belief POMDP using a varient of PERSEUS in which the CBG solver is used to compute maximizations.

ORIGINALITY:

To my knowledge, the integer linear programming formulation of the CBG is novel, as is the optimal belief compression scheme. The latter in particular is a very original and intriguing idea. The PBVI method is also novel as far as I know, though it is only a minor variation on PERSEUS. However, I have serious doubts about the originality of the belief POMDP formulation presented in Section 4. In particular, it bears a striking similarity to the continuous-state MDP approach recently proposed by Dibangoye et al. [1]. Though those authors use an MDP formulation, this is really just the belief MDP of a POMDP that, as far as I can tell, is conceptually identical to the belief POMDP formulation used in this paper. In Dibangoye et al., the actions correspond to assignments of decision rules to each agent, such that the central planner has a Markov state representation but each decision rule conditions only on individual observations available to each agent. This is exactly the approach taken here as well. Section 4 may also relate to recent work by Oliehoek [2], though the exact nature of the relationship is less clear to me in that case. The fact that this submission doesn't compare to or even cite these highly related papers is a significant concern for me.

CLARITY:

Overall this is a polished and well written work. I found some aspects of the exposition unclear (see detailed comments below) but these were not fatal.

SIGNIFICANCE:

Given the questions about novelty of Section 4, the algorithmic contribution is moderate: somewhat incremental but certainly nontrivial. In my opinion, the most interesting and potentially significant aspect is the belief compression approach of Section 5. However, I am also somewhat skeptical that this idea is practical. Intuitively, it seems that there must be a trade-off: asking the planner to determine how best to compress beliefs must increase computational costs that might or might not outweigh the benefit of compressing beliefs. Unfortunately, no analysis (or even mention) of this trade-off appears in the paper. Furthermore, the empirical analysis does not isolate the effect of this component of the algorithm. This is an example of a more general problem limiting the paper's significance. Because the experiments only assess the complete system, no insights are gleaned into how each component contributes to performance. Does it work because of or in spite of each component? Is the total performance gain more than the sum of that of its parts? The paper would be greatly improved if it addressed such questions.

Detailed comments:

1: PERSUES -> PERSEUS

3: The explanation of the integer linear program is unclear, which I believe is due to confusion between state and joint observations. The variable x_a\theta is indexed by \theta but then we are told there is one for each state. (2) contains no mention of state but the text below it refers to feasible assignments of x_as. Also, constraints 2 -> constraints (2).

4: I don't understand why the assumption there is a bounded number of possible beliefs is important. First of all, it seems to me that in any finite Dec-POMDP, there is at any timestep only a finite number of reachable beliefs. The fact that this number can increase over time would mean that the size of the action space of the belief POMDP grows over time, but I don't see why that's problematic for the formulation. I can understand bounding the number of beliefs as a compression technique to yield efficient approximations, but this bound is presented as a prerequisite for the conversion to a belief POMDP which doesn't seem right to me. Also, the use of the word "observation" is confusing in this section. How can there be a one-to-one mapping between observations and beliefs? Do you mean observation history? Similarly, we are told on line 173 that there are no observations in the belief POMDP but then told in line 178 that actions in this belief POMDP are mappings from observations to actions. Why is the Lemma numbered 4.0.1?

5: Is t_i the same as |B|?

6: What is a concave and locally linear value vector? Aren't these vectors piecewise linear and convex?

7: To what horizon where these problems tested? How many hours were used by the methods that obtained the "Previous Best Utility"? If it's not the same as for your PBVI method, how can the comparison be fair?

Refs: Is [18] correct? It has the same title and authors as [17].


[1] Optimally Solving Dec-POMDPs as Continuous-State MDPs. Jilles S. Dibangoye, Christopher Amato, Olivier Buffet and François Charpillet. In Proceedings of the Twenty-Third International Joint Conference on Artificial Intelligence (IJCAI-13), 2013.

[2] Frans A. Oliehoek. Sufficient Plan-Time Statistics for Decentralized POMDPs. In Proceedings of the Twenty-Third International Joint Conference on Artificial Intelligence, 2013.
Summary: An algorithm with several components is proposed: one component has doubtful novelty given recent work that is not cited here. Another component is original and interesting but its practicality is unclear. The experiments do not validate individual components but only the whole, leaving important questions unanswered.

Submitted by Assigned_Reviewer_5

The paper describes a new approach for DEC-POMDPs. Three contributions are made:

* An approach to convert DEC-POMDPs to bounded belief DEC-POMDPs
* An approach to convert bounded belief DEC-POMDPs to POMDPs with exponentially many actions
* An integer linear program to optimize one-step look-ahead policies in POMDPs with exponentially many actions

These are significant and original contributions. Their combination results in a new algorithm that found the best policies so far for a set of DEC-POMDP benchmarks.

The paper is difficult to read, however I believe this is because of a lack of space. Each of the 3 contributions could be a paper on its own. As a result, the paper describes concepts at a high level, but many details are missing. If the paper is accepted, I recommend expanding the paper by including more details for each approach in an appendix. This will help readers to implement the approaches and build on this work.

The empirical evaluation is adequate. However there is one important piece of information missing. The running times of the different algorithms are not reported. It is great that the proposed algorithm finds better policies than previous methods, but the reader is left wondering whether this is simply because the proposed algorithm used more time than previous algorithms. Please include a comparison of the running times.

The belief compression technique is very interesting. I'm wondering what are the links between this belief compression technique and finite state controllers. In the proposed compression technique, the agents decide for themselves how to aggregate beliefs. Something similar happens in controllers. Consider a controller with different nodes at each time step such that nodes at time t can only transition to nodes at time t+1. Controllers are parameterized by two mappings: an action mapping (n->a) that selects the action to be executed in each node and a next node mapping (n,o->n') that indicates which node n' will be reached from n after receiving observation o. The optimization of a controller involves the optimization of both mappings. In particular, the optimization of the next node mapping corresponds to the optimization of the aggregation of beliefs into a fixed number of beliefs at each time step. The proposed algorithm to optimize belief aggregation is based on value iteration while controller optimization is typically based on some form of policy search. So would it be fair to say that the algorithms are different but the end result is the same? It seems that I could take the fixed set of aggregated beliefs at each step and construct a corresponding set of nodes. The optimized mapping from beliefs-observation pairs to aggregated beliefs corresponds to the next node mapping in controllers.

Summary: This is excellent work that makes three important contributions. The result is a new algorithm for DEC-POMDPs that finds better policies than previous methods for a set of benchmark problems.

Submitted by Assigned_Reviewer_6

The paper presents four contributions to the solution of dec-POMDPs: a novel integer program for solving Bayesian games, followed by a belief compression method for dec-POMDPs that converts them into normal POMDPs (under the assumption of a fixed and known number of beliefs for each agent at each time step). Finally, an adaptation of a point-based value iteration technique that allows the POMDPs thus constructed to be solved.

The one thing that I found confusing about the paper is the use of "dynamics types" in the experiments. It was not very clear what this meant, or where I could find this in the algorithm development. It seems this are the beliefs of each agent, and thus are the free parameters t_i referred to in section 5 (the number of beliefs each agent uses to represent each other agent). However, these are then referred to as \theta, type, belief factor, and simply agent belief in other parts of the paper. This could be clarified. I was also unclear on why the BB-DecPOMDP gets the belief factor \theta as an observation at each time step. This leaves me a bit concerned that this method is somehow benefitting from this additional information in ways that the other methods compared to are not. It would be good to have a statement about the baseline methods that are compared against, so readers can be more confident that these comparisons are fair.

The results look very good, although the missing Mars Rover problems are a significant omission. How much memory would be required to solve these problems? It seems 8Gb is not terribly much these days - could the authors not attempt this on a larger machine or cluster?
Summary: the paper presents a number of contributions to the dec-POMDP literature, and the results look good. A few clarity issues could be addressed and the omission of a significant benchmark problem is glaring.
Author Feedback

Author rebuttal: We would like to thank the reviewers for their thoughtful reviews. Our response follows:

Reviewer 1)

Occupancy MDPs (OMDPs) [1] are indeed very similar to belief-POMDPs. At a fixed timestep, they look identical; however, an OMDP is a finite horizon model and a belief-POMDP is an infinite horizon model. To retain a tractable model we assume the number of beliefs do not grow indefinitely over time. Belief-POMDPs are able to make use of this assumption because of a key non-trivial insight: We do not need to remember which histories correspond to which beliefs; the common-knowledge distribution over states is sufficient. In fact, the mapping from histories to beliefs changes over the course of the algorithm. This allows the same set of beliefs to be used at each timestep, and thus allows us to compute a single [finite] PWLC value function that is valid for all time-steps of the problem (i.e. can be used for an infinite horizon policy).

Without recognizing that beliefs can be relabeled we would have to create a separate value function for each time-step (as in [1]). Without a bounded number of beliefs, our belief-POMDP would have an infinite number of underlying states, and would not be suitable for traditional POMDP methods. Dibangoye et.al. [1] can ignore this issue because a finite horizon problem inherently has bounded belief (albeit exponential in the horizon). In belief-POMDPs we can use the same set of beliefs at each timestep, allowing us to compute a [finite] PWLC value function valid for infinite horizon policies.

Oliehoek [2] is also very close to developing a model similar to OMDPs. Unfortunately, we were made aware of [1] and [2] after the NIPS deadline: while preprint versions of these papers are available on the authors website, both papers are to appear in IJCAI’13 which only released their proceedings a few days ago (Aug. 2). In light of these papers, we would connect belief-POMDPs to [1] and [2] while leaving our remaining three results intact. We are still confident that these three results are substantial, novel, and impactful. In particular, we believe our optimal belief compression scheme is a major breakthrough and the most important contribution of this paper.


Reviewer 1 & 3)

> “How can there be a one-to-one mapping between observations and beliefs? “

This is probably the most important point of confusion for us to address (and is related to our response above). There is a difference between a belief that is a true sufficient statistic (no worse than knowing the full history) and a compressed-belief. When we say that beliefs are bounded, we mean so if the first sense. We therefore know a priori the mapping from action-observation-histories to beliefs (because we know which histories are information-equivalent). Using induction, (with a base case of the first observation), if we assume the previous observation was the previous belief, we can replace each agent’s [previous-belief, observation] pair with a new observation (at most m) that corresponds to the new belief. Because we know this transformation a priori, we can fold it into the transition function. Therefore, in the BB-DecPOMDP, beliefs and observations are synonymous. During our optimal belief compression scheme it is the assumption that beliefs in the BB-DecPOMDP are sufficient (so that agents don’t need to remember history) that is the true point of approximation.

In the belief-POMDP the centralized planner gets no observations; however, the actions he gives are local decision rules for each agent (using the terminology from [1]) which are mappings from local-observations to actions.

> To what horizon were these problems tested?

We are addressing the infinite horizon problem so we ran our algorithm until convergence and simulated the resulting policy to a horizon that had an error below the reported significant digits. This was a horizon of about 1500 for the Wireless problem (gamma = .99) and 150 for the others (gamma = .9). The other algorithms that we compare against reported results likewise.


Reviewer 2)

> “what are the links between this belief compression technique and finite state controllers?”

A finite state controller doesn’t include knowledge about the common-knowledge distribution (the state of the belief-POMDP), which is continuous and thus not finite. In BB-DecPOMDPs, actions and therefore belief transitions can depend on this distribution. This additional information makes the optimization problem convex and thus we don’t suffer from local optima like most FSC algorithms.

For a fixed horizon and starting belief, a FSC with a width of m (# beliefs) states at each time-step could be constructed that produces the same policy as us for that fixed horizon. Such a FSM could not be turned into an infinite horizon policy as there might be an unbounded number of distinct beliefs over time. For example, consider dec-tiger with the modification that if the tiger is behind the left door then when the game resets the tiger stays behind the left door. For this example, each time the game resets it becomes more and more likely that the tiger is behind the left door but the probability will never converge to 1.0. A finite value-function approach could have value-vector changes (and thus policy changes) arbitrarily far in the future (close to 1.0).


Reviewer 3)

> It seems these are the beliefs ...

Your understanding is correct, and we acknowledge we should be more consistent with terminology.

> This leaves me a bit concerned...

Hopefully the “one-to-one mapping” response above already convinced you, but if not, we also verified that information leaking wasn’t happening by simulating our computed policies on the original DecPOMDP (where agents can only base their actions on their own observations). We compared the simulated expected value against the value function and found them to be equal up to our convergence tolerance.